# Inhibitory Potential of Polyclonal Camel Antibodies against New Delhi Metallo-β-lactamase-1 (NDM-1)

**DOI:** 10.3390/molecules25194453

**Published:** 2020-09-28

**Authors:** Rahma Ben Abderrazek, Sarra Chammam, Ayoub Ksouri, Mariagrazia Perilli, Sayda Dhaouadi, Ines Mdini, Zakaria Benlasfar, Gianfranco Amicosante, Balkiss Bouhaouala-Zahar, Alessandra Piccirilli

**Affiliations:** 1Laboratoire des Venins et Biomolécules Thérapeutiques, Institut Pasteur Tunis, Université Tunis El Manar, 13 Place Pasteur, BP-74, 1002 Tunis, Tunisie; ch.sarra@gmail.com (S.C.); ayoub.ksouri@pasteur.utm.tn (A.K.); sayda.dhaouadi@hotmail.fr (S.D.); inesmdini@gmail.com (I.M.); zbconsultingvet@hotmail.com (Z.B.); balkiss.bouhaouala@fmt.utm.tn (B.B.-Z.); 2Dipartimento di Scienze Cliniche Applicate e Biotecnologiche, Università degli Studi dell’Aquila, Via Vetoio, I-67100 L’Aquila, Italy; mariagrazia.perilli@univaq.it (M.P.); gianfranco.amicosante@univaq.it (G.A.); alessandra.piccirilli@univaq.it (A.P.); 3Faculté de Médecine de Tunis, Université Tunis El Manar, 15 Rue Djabel Lakhdar, 1007 Tunis, Tunisie

**Keywords:** potential inhibitors, antibiotic resistance, metallo-β-lactamases, NDM-1, camel antibodies

## Abstract

New Delhi Metallo-β-lactamase-1 (NDM-1) is the most prevalent type of metallo-β-lactamase, able to hydrolyze almost all antibiotics of the β-lactam group, leading to multidrug-resistant bacteria. To date, there are no clinically relevant inhibitors to fight NDM-1. The use of dromedary polyclonal antibody inhibitors against NDM-1 represents a promising new class of molecules with inhibitory activity. In the current study, immunoreactivities of dromedary Immunoglobulin G (IgG) isotypes containing heavy-chain and conventional antibodies were tested after successful immunization of dromedary using increasing amounts of the recombinant NDM-1 enzyme. Inhibition kinetic assays, performed using a spectrophotometric method with nitrocefin as a reporter substrate, demonstrated that IgG1, IgG2, and IgG3 were able to inhibit not only the hydrolytic activity of NDM-1 but also Verona integron-encoded metallo-β-lactamase (VIM-1) (subclass B1) and L1 metallo-β-lactamase (L1) (subclass B3) with inhibitory concentration (IC_50)_ values ranging from 100 to 0.04 μM. Investigations on the ability of IgG subclasses to reduce the growth of recombinant *Escherichia coli* BL21(DE3)/codon plus cells containing the recombinant plasmid expressing NDM-1, L1, or VIM-1 showed that the addition of IgGs (4 and 8 mg/L) to the cell culture was unable to restore the susceptibility of carbapenems. Interestingly, IgGs were able to interact with NDM-1, L1, and VIM-1 when tested on the periplasm extract of each cultured strain. The inhibitory concentration was in the micromolar range for all β-lactams tested. A visualization of the 3D structural basis using the three enzyme Protein Data Bank (PDB) files supports preliminarily the recorded inhibition of the three MBLs.

## 1. Introduction

Infectious diseases caused by Gram-negative bacteria are most certainly the cause of morbidity and mortality worldwide [1,2]. β-Lactam antibiotics are the most common antimicrobials used in clinical therapy, but the extensive use of these molecules has led microbes to develop new mechanisms of resistance. The production of β-lactamases represents the most prevalent bacterial-resistance mechanism against β-lactam antibiotics, and it is now a major concern throughout the world [3,4]. To date, more than 2700 different β-lactamases exhibiting a wide range of primary structures and catalytic properties have been described [3]. On the basis of their amino-acid sequences, β-lactamases are categorized into four molecular classes, A, B, C, and D [5]. Classes A, C, and D are serine-β-lactamases, while class B metallo-β-lactamases (MBLs) are metallo-enzymes requiring one or two zinc ions for their activity. According to amino-acid sequence identity and zinc ion dependence, MBLs are classified into three subclasses (B1, B2, and B3). The B1 subclass includes most clinically significant enzymes able to degrade all classes of β-lactams, except monobactams, but with a special activity toward carbapenems [6,7]. The MBLs are especially alarming due to (1) their potential for horizontal transfer, (2) their large activity profiles that encompass most β-lactam antibiotics, and (3) the absence of clinically useful inhibitors. Indeed, they are not sensitive to β-lactamase inhibitors such as clavulanic acid, sulbactam, and tazobactam that are used in clinical therapy [8,9,10,11,12]. The most common MBLs are represented by Verona integron-encoded metallo-β-lactamase (VIM), imipenemase (IMP), and New Delhi metallo-β-lactamase-1 (NDM-1) [13,14,15]. The *bla*IMP-like and *bla*VIM-like genes have been identified in most clinically relevant bacteria belonging to the Enterobacteriaceae family, in *Pseudomonas* spp., and in *Acinetobacter* spp., while *bla*NDM-1 has mainly been found in Enterobacteriaceae. Among them, NDM-1 represents a worldwide threat to healthcare [16,17].

Due to the emergence of new enzyme markers, especially NDM-1, that hampered the capability of all antibiotics of the β-lactam group to treat infections caused by microorganisms carrying such resistances, the use of non-β-lactam inhibitors is one of the solutions. In this context, antibody inhibitors that can bind MBLs with high affinities seem to constitute a new and attractive approach.

Camelid immune systems produce conventional and heavy-chain homodimer antibodies (HcAbs) devoid of light chains [18]. These promising natural HcAbs bind with high affinity to their targets of interest, particularly those antigens that are unreactive for other immunoglobulins [19]. Camelid immunoglobulin G (IgG)was reported to be less immunogenic and more thermostable than IgG from other mammals [20]. The unique characteristic of the variable domain of HcAbs termed nanobodies [21,22] has sped up progress in antibody development against various antigens such as enzymes [23,24,25]. Several studies succeeded in developing antibody inhibitors targeting other β-lactamases, on the basis of camelid single-domain antibodies [26,27], which makes them potential candidates and an alternative treatment for infections caused by multidrug-resistant bacteria.

In this study, our main goal was to target NDM-1, the most prevalent type of metallo-β-lactamase responsible for multidrug-resistant bacteria using highly specific dromedary immunoglobulins. To achieve this, we immunized one dromedary with NDM-1. Then, specific immunoglobulin G (IgG) subclasses including conventional (IgG1) and heavy-chain antibodies (HC-IgG2, HC-IgG3) from immune serum were retrieved, purified, and characterized. The antigen immune reactivity of purified polyclonal antibodies was assessed. We tested the inhibition capacity of IgGs subclasses against NDM-1, as well as VIM-1 (subclass B1) and L1 metallo-β-lactamase (L1) (subclass B3) Moreover, an Antimicrobial Susceptibility Test was evaluated by minimum inhibitory concentration (MIC) determination using two β-lactams belonging to the family of carbapenems. Finally, molecular investigations of the metallo-β-lactamase structures (NDM1, L1, and VIM1) were carried out using the Protein data bank (PDB) crystal structures of NDM1, L1, and VIM1

## 2. Results

### 2.1. Determination of K_m_ for NDM-1, VIM-1, and L1

Recombinant NDM-1 enzyme with a degree of purity higher than 98% (data not shown) was used to immunize a female dromedary and to determine the *K_m_* value for nitrocefin. Nitrocefin *K_m_* values of 15 μM ± 3, 17 μM ± 1, and 10 μM ± 1 were calculated for NDM-1, VIM-1, and L1 metallo-β-lactamases, respectively.

### 2.2. Humoral Immune Response Elicited in the Dromedary

To generate polyclonal dromedary antibodies with high inhibitory potential, we hyperimmunized a female dromedary aged 10 years old with increasing amounts of recombinant protein NDM-1. During the first three subcutaneous injections, the dromedary received increasing amounts of NDM-1 ranging from 250 to 350 μg/injection followed by three boosters of 400 μg of NDM-1. During the immunization, the dromedary serum titer (diluted at 1:5000 in Phosphate-Buffered Saline (PBS) was recorded by ELISA, showing a high antigen-specific response induced after the fourth injection (day 21). This response was kept high after boosts with recombinant NDM-1 (Figure 1).

Absorbance values registered for Bovine Serum Albumin protein, incubated as an irrelevant control with sera from days 0 and 49, did not exceed 0.3 optical density units/ mL (ODU/mL) which proved the specific reactivity of the immune serum to NDM-1 enzyme. After immunization, dromedary serum was taken and fractionated into conventional and HC-IgG isotypes. The immune reactivity of these fractions confirmed the presence of specific dromedary raised antibodies against recombinant NDM-1 compared to the preimmune serum used as a negative control (Figure 2).

### 2.3. IgG Subclass Characterization

Three IgG subclasses were isolated by affinity chromatography on protein A and G columns including conventional antibodies and two heavy-chain antibodies, as illustrated by SDS-PAGE (Figure 3). The first eluted fraction at pH 3.5 through the protein G affinity column (IgG3) corresponded to a heavy-chain antibody with an H-chain molecular weight of 43 kDa (lane 3), whereas another heavy-chain antibody (IgG2) was bound on the protein A column and purified at pH 4.5 with an H-chain molecular weight of 46 kDa (lane 2). The only conventional antibody purified from protein G at pH 2.7 had a molecular weight of 50 kDa (H-chain) and 25 kDa (L-chain), as indicated in lane 1. All of SDS-PAGE analyses were performed under reducing conditions.

The proportion of the different IgG fractions purified from 2 mL of hyperimmunized dromedary serum is reported in Table 1. As expected, the best proportion was attributed to the IgG1 fraction with a concentration of 0.7 mg/mL. IgG2 and IgG3 fraction concentrations were 0.17 mg/mL and 0.3 mg/mL, respectively. Moreover, the proportion of the three purified IgG fractions was 29.25%. The most abundant fraction was IgG1, which represented 59.82% of total IgGs, whereas IgG2 and IgG3 represented 14.53% and 25.64%, respectively. Thus, HCAbs represented 40.17% of total IgGs.

### 2.4. Inhibition Kinetic Assays of NDM-1

Fixed concentrations (3000 nM) of antibodies IgG1, IgG2, and IgG3 were preincubated with 75 nM of NDM-1m and the enzyme activity was measured by following the hydrolysis of 60 μM nitrocefin. As shown in Figure 4A, the inhibition of NDM-1 is time-dependent. Interestingly, IgG2 and IgG3 were able to inhibit 30% and 40% of the NDM-1 activity after 60 min of preincubation. However, IgG1 inhibited 50% of NDM-1 activity only after 1 min of preincubation. In order to calculate the inhibitory concentration IC_50_ values for the three IgGs, kinetic experiments were performed preincubating NDM-1 with increasing concentrations of IgG1 and IgG2/IgG3 for 1 and 60 min, respectively (Figure 4B–D). After 1 min of pre-incubation with NDM-1, IgG1 was able to strongly inhibit the enzyme (Figure 4B), and the IC_50_ was calculated to be 0.45 μM (Table 2). NDM-1 and IgG3 were preincubated for 60 min, and the nitrocefin hydrolysis was followed at increasing concentration of IgG3 (from 0 to 100 μM) (Figure 4D). The IC_50_ was calculated to be 16 μM (Table 2). On the contrary, increasing the concentration of IgG2 to 100 μM, after 60 min of preincubation with NDM-1, was unable to reduce the enzyme activity below 80%. Thus, IC_50_ was estimated to be >100 μM. Using rabbit IgG as a negative control, the NDM-1 enzyme was found to preserve 100% of residual activity.

### 2.5. Inhibition Kinetic Assay of VIM-1 and L1 Enzymes

The inhibition activity of NDM-1-specific IgG1, IgG2, and IgG3 was also tested against two other metallo-β-lactamases, VIM-1 and L1. Concerning VIM-1, IgG1, IgG2, and IgG3 inhibited 50% of enzyme activity after 1 min of preincubation (enzyme + antibody) (Figure 5A).

Using an increasing concentration of IgGs, it was possible to calculate the IC_50_ (Figure 5B and Table 2), the values of which were lower than that calculated for NDM-1, particularly IgG2, with IC_50_ values of 0.04 and 1 μM for VIM-1 and L1, respectively (Table 2).

Similar behavior was observed for the L1 metallo-β-lactamase. IgG1, IgG2, and IgG3 were able to inhibit 50% of L1 activity after 1 min (IgG1 and IgG3) and 5 min (IgG2) of incubation (Figure 5C). The IC_50_ values calculated for the three antibodies (Figure 5D and Table 2) were lower than that of NDM-1. Using rabbit IgG as a negative control, VIM-1 and L1 MBLs were found to preserve 100% of residual activity.

### 2.6. Antimicrobial Susceptibility Test

The phenotypic profile of the strains was evaluated by MIC determination using two β-lactams belonging to the family of carbapenems (imipenem and meropenem). All strains tested, with the exception of *Escherichia coli* BL21(DE3)CodonPlus/pET-24, were resistant to imipenem and meropenem with MIC values >64 mg/L. The addition of 4 and 8 mg/L IgGs was unable to restore the susceptibility of meropenem and imipenem.

### 2.7. Periplasm Extraction and Specific Activity Determination

In order to evaluate the presence of β-lactamases in the periplasm extract, cultures of *E. coli* BL21(DE3)codonplus/pET-24/blaNDM-1, *E. coli* BL21(DE3)codonplus/pET-24/blaVIM-1, *E. coli* BL21(DE3)codonplus/pET-24/blaL1, and *E. coli* BL21(DE3)codonplus/pET-24 strains were prepared. Each periplasm was used to determine specific activity of NDM-1, VIM-1, and L1 and their IC_50_ with IgGs. The IC_50_ calculated using the periplasm of each strain gave the same values as for pure recombinant enzymes, ranging from 100 to 0.036 µM (Table 2).

### 2.8. Molecular Investigation

Molecular investigations of the metallo-β-lactamase structures (NDM1, L1, and VIM1) were carried out using the Protein Data Bank (PDB) crystal structures of NDM1, L1, and VIM1 (3SPE, 1SML, and 5N5G, respectively). The molecular representations of enzyme structures and electrostatic potentials were visualized using PyMOL. As shown in Figure 6B, NDM1, L1, and VIM1 exhibit substantial three-dimensional (3D) structural similarity. The shapes visualized according to the rainbow coloring show the conserved α-helix-rich 3D general topology with limited divergence in the overall structure. Interestingly, the focus on the topology of the catalytic site of each enzyme shows a common concave pocket within the three enzymes (Figure 6C). More interestingly, the investigation of the surface electrostatic potentials of the enzymes showed some divergence in the surface charges of catalytic sites, which may support the large inhibition activity of NDM-1-specific IgG1, IgG2, and IgG3 (Figure 6A). The conserved amino acids forming a conserved structural motif between the three enzymes were as follows: L1: H84, H86, H160, D88, H89, and H225; VIM-1: H88, H90, H153, D92, C172, and H214; NDM-1: H120, H122, H189, D124, C208, and H250.

## 3. Discussion

NDM-1 was found to confer enteric bacteria resistance to nearly all β-lactams, including the heralded carbapenems, causing a serious threat to human health around the world [28,29]. At the present time, commercial metallo-β-lactamase inhibitors are not available for the treatment of common and serious infections. Thus, the development of new NDM-1-specific binders constitutes a challenge of the current time.

The camelid immune system is unique amongst mammals, whereby over 50% of the serum immunoglobulins are IgG heavy-chain-only antibodies [30]. Their small antigen-binding fragment, called a nanobody (15 kD), was proven to have a functional recognition which interacts with MBL enzymes [27]. To our best knowledge, there are no studies related to the inhibitory assay determination of camel antibodies against NDM-1. In the present study, a new approach incorporating dromedary immunoglobulins using NDM-1 enzyme was established.

On the basis of the ELISA results, the immunization of the dromedary using increasing amounts of NDM-1 stimulated the production of polyclonal antibodies. NDM-1 is immunogenic, and it elicits a humoral response and specific antibody isotypes. Indeed, all dromedary IgG subclasses containing HCAbs and conventional antibodies were reactive to NDM-1 enzyme in an antigen-specific binding manner. In addition, the quantity of HCAbs exceeded 40%. This proportion is in accordance with the proportions of HCAbs reported in the literature, which represent up to 50% of total IgGs [30].

The polyclonal antibodies were tested against NDM-1, as well as VIM-1 and L1 β-lactamases. Interestingly, NDM-1-specific IgGs are also able to inhibit VIM-1 and L1. Our data clearly demonstrated that NDM-1-specific IgG1, IgG2, and IgG3 were able to inhibit VIM-1 with IC_50_ values (calculated for the three antibodies) lower than that seen for NDM-1. Indeed, the enzyme L1 of *Stenotrophomonas maltophilia* was better inhibited by the three IgGs with a similar IC_50_ value to VIM-1. The L1 metallo-β-lactamase is unique among β-lactamases with a tetrameric structure, and it is produced by *Stenotrophomonas maltophilia*, a nosocomial pathogen of immunocompromised patients (cancer, cystic fibrosis). These results could be explained in part by the polyclonal origin of the NDM-1-specific IgGs.

In vitro experiments using recombinant *E. coli* strains carrying NDM-1, VIM-1, and L1 showed that the IgGs were unable to restore the susceptibility of carbapenems. Interestingly, IgGs were able to inhibit the hydrolytic activity of the three enzymes when tested on the periplasm extract of each cultured strain. As observed with the pure recombinant enzymes, the IC_50_ values were in the micromolar range for all *β*-lactams tested in the periplasm extract. We deduced that IgGs are able to specifically inhibit the bacteria-expressed enzymes but are unable to translocate through the bacterial membrane. This interesting result could be explained by the high molecular weight of the IgGs, exceeding 100 kDa. Our “in-progress” investigations using a collection of MBL-positive strains will probably provide further explanation regarding the camel IgG inhibition mechanism.

Therefore, a visualization of the 3D structural basis, which preliminarily supports these results, was carried out using the three enzyme PDB files. NDM-1 and VIM-1 are enzymes belonging to subclass B1 metallo-β-lactamases, and they showed a low sequence identity (about 32%) despite generally similar 3D structures (Figure 6). The similar structure of the two enzymes could explain the efficacy of the NDM-1 polyclonal antibody inhibitors against VIM-1. However, the observed difference in the electrostatic potentials of the three enzyme surfaces indicated that the molecular interaction is more complex and not only supported by the surface electrostatic potentials. The active sites, located at the bottom of a shallow groove between loops L3 and L10, which play an important role in substrate hydrolysis, have a similar topology with few divergent amino-acid lateral chains. In both enzymes, two zinc ions are coordinated by H120, H122, and H189 (site Zn1), D124, C208, and H250 (site Zn2), and a water molecule (or a hydroxide moiety). Further structural studies on the basis of modeling and docking will be conducted as soon as the anti-MBL IgG amino-acid sequences are available.

In summary, the success obtained with antibodies derived from dromedary immunization with NDM-1 was well demonstrated in this study. Further experiments involving monoclonal camelid single-domain nanobodies, with minimal size (15 kDa) and high solubility, which bind to the enzyme with nanomolar affinity, will be needed in order to better explore the inhibition mechanism. With this approach, it seems possible to obtain different inhibitors of MBLs to fight antibiotic resistance, and we can expect new opportunities for the treatment of bacterial infections with existing antibiotics.

## 4. Materials and Methods

### 4.1. Enzymes, Plasmids, and Strains

VIM-1 [31] and L1 [32] metallo-β-lactamases were obtained from the Clinical Biochemistry and Molecular Biology Laboratories of the University of L’Aquila. The gene *bla*_NDM-1_ was cloned without a signal peptide in the pET-24(a) vector using the *Nde*I and *Xho*I restriction sites to obtain plasmids pFM-NDM-1 [33,34]. *E. coli* Novablue competent cells (*endA1 hsdR17*(r_K12_^−^ m_K12_^+^) *supE44 thi-1 recA1 gyrA96 relA1 lac* F[*proA^+^B^+^ lacI^q^Z*Δ*M15*:Tn*10*] (Tet^R^)) were used for the initial cloning as a nonexpression host. The recombinant plasmid was then transferred into *E. coli* BL21(DE3)codon-plus (*E. coliB F-ompT hsdS(rB^−^mB^−^)dcm+Te trgalλ(DE3)endAHte [argU proLCamr*]) for enzyme expression. The authenticity of cloned mutant genes was verified by sequencing both strands of the three recombinant plasmids using an automated sequencer (ABI PRISM3500, Life Technologies, Thermo fisher Scientific, Monza, Italy).

### 4.2. Production and Purification of NDM-1 Metallo-β-lactamase

*E. coli* BL21(DE3) codon plus cells containing the recombinant pFM-NDM-1 plasmids were grown in 1 L of Tryptic Soy Broth medium with 50 μg/mL kanamycin at 37 °C in an orbital shaker (180 rpm). Each culture was grown to achieve an A_600_ of approximately 0.5, and 0.4 mM IPTG (isopropyl-β-thiogalactoside) was added. After the addition of IPTG, the cultures were incubated for 16 h at 22 °C, under aerobic conditions. Purification of the NDM-1 enzyme was performed using the protocol reported in our previously published study [35].

### 4.3. Camel Immunization Protocol

A female dromedary (*Camelus dromedarius*) was used for the antibody production against NDM-1. All procedures concerning the dromedary immunization protocols were approved by the Ethical Committee of Pasteur Institute of Tunis (acceptance reference: 46/19). Before the first injection, preimmune sera were collected, and the dromedary received subcutaneous doses ranging from 250 μg to 400 µg of NDM-1 in physiological saline solution (PSS) via subcutaneous (s.c.) injection at weekly and biweekly intervals (days 0, 7, 14, 21, 35, and 49). In total, the animal received 2 mg of enzyme. In the first injection, complete Freund’s adjuvant (Sigma-Aldrich) was used. The first injection was performed using an equal volume of NDM-1 and complete Freund’s adjuvant, while booster injections were administered in incomplete Freund’s adjuvant emulsions as previously described [36]. Four days after the last injection (day 53), a blood sample was taken, and the serum was separated and stored at −20 °C.

### 4.4. IgG Subclass Fractionation

IgG subclasses were obtained via differential adsorption on Hitrap-protein A and G (Qiagen) columns as previously described [37]. First, 2 mL of immune camel serum was applied onto the Protein G column. After washing with 20 mM PBS (pH 7.2), IgG3 and IgG1 were successfully bound to the Hitrap-protein G column, while the IgG2 subclass was not adsorbed. IgG3 was eluted with 0.15 M NaCl and 0.58% acetic acid, pH 3.5. The IgG1 fraction was eluted with 0.1 M glycine-HCl, pH 2.7. The unadsorbed IgG2 fraction was loaded onto a protein A column and subsequently eluted with 0.15 M NaCl and 0.58% acetic acid buffer, pH 4.5. All the purified IgG subclasses were immediately neutralized using 1 M Tris HCl, pH 9.0. IgG concentrations were obtained using a Bicinchoninic Acid detection kit (Sigma). The purity of IgG fractions was determined on 12% SDS-PAGE under reducing conditions.

### 4.5. ELISA Assessment

The sera taken during the immunization program and the various amounts of the polyclonal IgG fractions were tested for a humoral immune response by ELISA evaluation [36,38]. Briefly, a 96-well Maxisorb plate was coated with NDM-1 (100 ng per well diluted in 100 μL of PBS buffer). After overnight incubation at 4 °C, residual protein binding sites were blocked with 5% BSA in PBS solution for 1 h at room temperature (RT). After several washing steps with PBS-T (PBS with 0.1% Tween-20), sera taken before each injection during the immunization program (diluted at 1:5000 in PBS) and various concentrations of IgG fractions (0, 2, 4, 8, and 10 µg/mL) were added to duplicated wells and incubated for 1 h at RT. To remove nonspecific antigen–antibody binding, the wells were extensively washed with PBS-T, and the subsequently bound camel sera and IgGs were detected using rabbit anti-camel antibodies (diluted 1:8000, Sigma); detection of rabbit antiserum was performed with mouse anti-rabbit antibodies conjugated to horseradish peroxidase (HRP) (Sigma) diluted 1:1000 in PBS. Finally, the enzymatic substrate tetramethylbenzidine (TMB) (sigma) was added. The reaction was stopped after 20 min with 50 μL of 2 N sulfuric acid, and absorption was measured at 450 nm.

### 4.6. Determination of K_m_ for NDM-1, VIM-1, and L1

Steady-state kinetic experiments were performed at 25 °C in 20 mM 4-(2-hydroxyethyl)-1-piperazineethanesulfonic acid HEPES, pH 7.0 + 20 µM ZnCl_2_ buffer to calculate *K_m_* values for NDM-1, VIM-1, and L1 using nitrocefin as a substrate [39]. Nitrocefin is a chromogenic cephalosporin (∆ε_482_ = 15,000 M^−1^·cm^−1^). Kinetic parameters were determined under initial-rate conditions using GraphPad Prism6 software to generate Michaelis–Menten curves. Data were collected with a Perkin-Elmer Lambda 25 spectrophotometer (Perkin-Elmer Italia, Monza, Italy). Each kinetic value was the mean of three different measurements; the error was below 1%.

### 4.7. Time-Dependent Inhibition Assays

The interaction of IgG1, IgG2, and IgG3 antibodies with NDM-1, VIM-1 and L1 was studied by kinetic assays using 60 μM nitrocefin as a reporter substrate [39]. Time-dependent inhibition assays were performed as described by Copeland et al. [40,41] by incubating 75 nM of each enzyme with a fixed concentration of antibodies (3000 nM) at 30 °C for 1, 5, 10, 20, 30, and 60 min. In order to determine the IC_50_ value, the enzymes (at a concentration of 75 nM) were made to react with increasing concentrations of IgG1, IgG2, and IgG3. Each kinetic value was the mean of three different measurements; the error was below 2%. Rabbit IgG (Sigma Aldrich, Milan, Italy) was used as a negative control in the kinetic inhibition assays at the maximum concentration of 5 μM.

### 4.8. Antimicrobial Susceptibility Test

The phenotypic profile (MIC) was characterized using the microdilution method with a bacterial inoculum of 5 × 10^5^ colony-forming unit /mLCFU/mL according to Clinical and Laboratory Standards Institute (CLSI) performance standards [41]. The IgGs were used in combination with imipenem and meropenem at fixed concentrations of 4 and 8 mg/L. The strains used for MIC determination were *E. coli* BL21(DE3)codonplus/pET-24/blaNDM-1, *E. coli* BL21(DE3)codonplus/pET-24/blaVIM-1, and *E. coli* BL21(DE3)codonplus/pET-24/blaL1. The *E. coli* BL21(DE3)CodonPlus/pET-24 strain was used as a control.

### 4.9. Periplasm Extraction and Specific Activity Determination

Periplasms were extracted from 10 mL of an overnight culture of *E. coli* BL21(DE3)codonplus/pET-24/blaNDM-1, *E. coli* BL21(DE3)codonplus/pET-24/blaVIM-1, and *E. coli* BL21(DE3)codonplus/pET-24/blaL1. The *E. coli* BL21(DE3)codonplus/pET-24 strain was used as a negative control. Cells were grown in TSB medium with 50 μg/mL kanamycin at 37 °C in an orbital shaker (180 rpm). Each culture was grown to achieve an A_600_ of approximately 0.5, and 0.4 mM IPTG (isopropyl-β-thiogalactoside) was added. After the addition of IPTG, the cultures were incubated for 16 h at 22 °C, under aerobic conditions. Cells were harvested by centrifugation at 8000 rpm for 10 min at 4 °C, and then washed twice with 25 mM sodium phosphate buffer (pH 7.0). The cells were then suspended in 30 mM Tris-HCl buffer (pH 8.0) containing 27% sucrose. The periplasm fraction was extracted via the addition of lysozyme (final concentration, 0.4 mg/mL) to the cooled solution. After 50 min of incubation on ice, the reaction was ended by adding MgSO_4_ (final concentration, 5 mM). The sample was then centrifuged at 30,000 rpm for 30 min (4 °C). The supernatant was dialyzed overnight at 4 °C in 20 mM HEPES, 20 µM ZnCl_2_ buffer with pH 7.0. Specific activity was measured using a spectrophotometric assay following the hydrolysis of 100 μM nitrocefin (λ = 482 nm; (∆ε_482_ = 15,000 M^−1^·cm^−1^), where one unit of β-lactamase activity was defined as the amount of enzyme which hydrolyzes 1 μmol of substrate per min at 25 °C in 20 mM HEPES, 20 µM ZnCl_2_ buffer with pH 7.0.

### 4.10. Molecular Visualization

The crystal structures of β-lactamases molecules NDM1, L1, and VIM1 were extracted from the Protein Data Bank (PDB) [42] using codes 3SPE, 1SML, and 5N5G, respectively.

Molecular visualization of the enzymes and electrostatic coloring were done using the molecular graphics system PyMOL, version 1.3 r1 (Schrödinger, L.L.C. (2010)) [43].

## Figures and Tables

**Figure 1 molecules-25-04453-f001:**
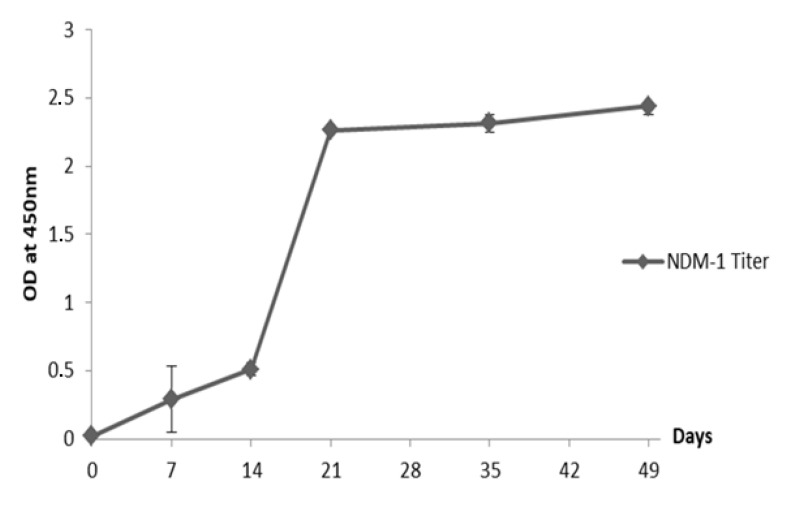
Reactivity of immune serum (1:5000) was tested by ELISA using New Delhi Metallo-β-lactamase-1 (NDM-1) at a concentration of 1 μg/mL. The absorbance values were measured at 450 nm. Error bars represent standard deviation.

**Figure 2 molecules-25-04453-f002:**
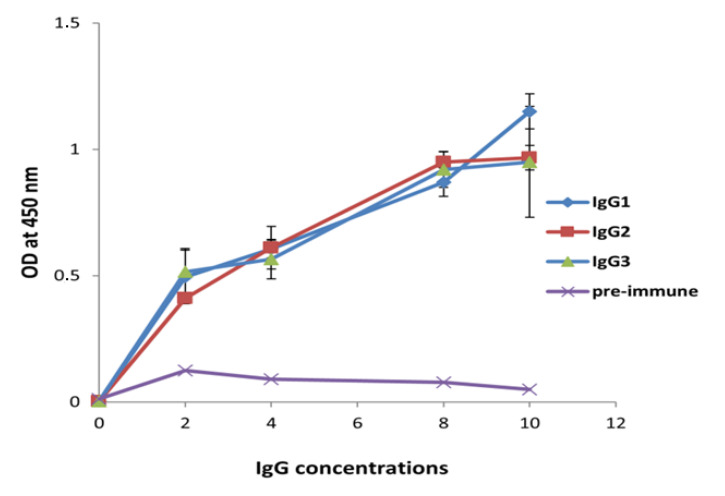
Reactivity of immunoglobulin G fractions was tested by ELISA using NDM-1 enzyme at a concentration of 1 μg/mL and various concentrations of IgG subclasses (µg/mL). Preimmune serum was used as a negative control. The signal values were measured at 450 nm. Error bars represent standard deviation.

**Figure 3 molecules-25-04453-f003:**
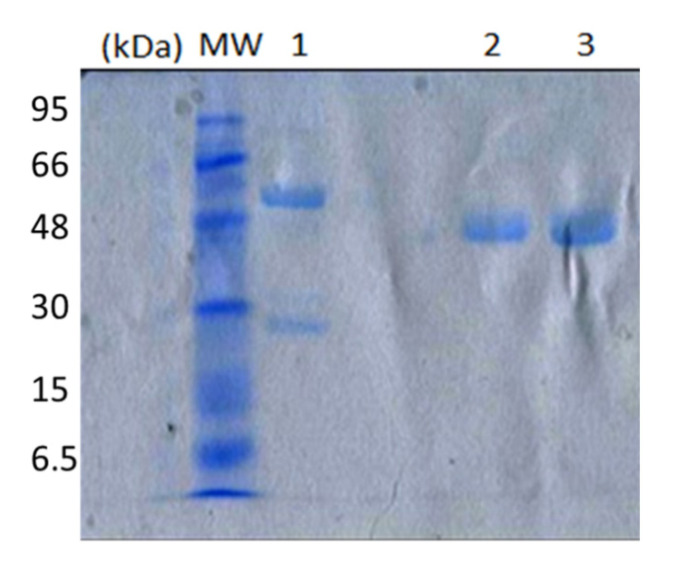
Analysis of purified IgG fractions on 12% SDS-PAGE under reducing conditions; lane 1, IgG1 with an apparent molecular weight of 50 kDa and 25 kDa. Lanes 2, 3 revealed a single band of 46 and 43 kDa corresponding to the heavy-chain antibodies (IgG2 and IgG3), respectively.

**Figure 4 molecules-25-04453-f004:**
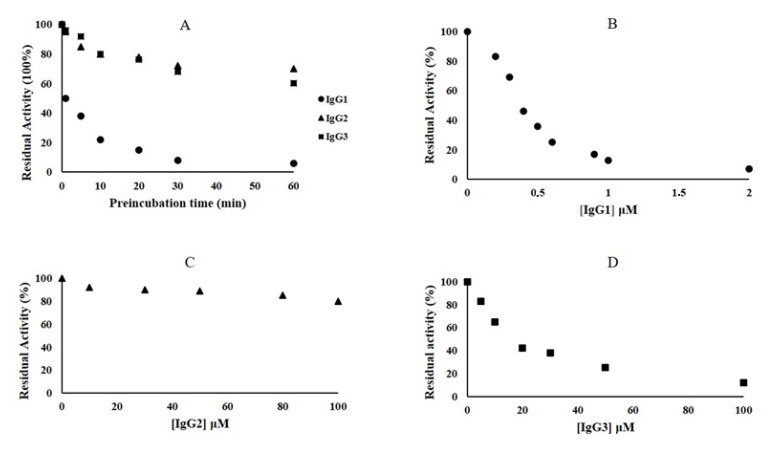
Inhibition effect of IgG1, IgG2, and IgG3 toward NDM-1. (**A**) Time-dependent inhibition assays of the three antibodies toward NDM-1; (**B**) residual activity (%) of NDM-1 after incubation (1 min) with increasing concentration of IgG1; (**C**) residual activity (%) of NDM-1 after incubation (60 min) with increasing concentration of IgG2; (**D**) residual activity (%) of NDM-1 after incubation (60 min) with increasing concentration of IgG3. Each kinetic value is the mean of three different measurements; the standard deviation (SD) was below 2%.

**Figure 5 molecules-25-04453-f005:**
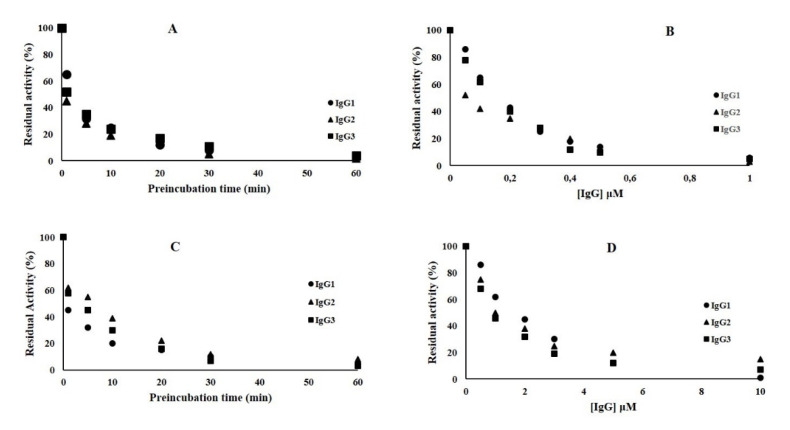
Inhibition effect of IgG1, IgG2, and IgG3 towardVerona integron-encoded metallo-β-lactamase VIM-1 and L1 metallo-β-lactamase L1. (**A**) Time-dependent inhibition assays of the three antibodies toward VIM-1 and L1; (**B**) residual activity (%) of VIM-1 after preincubation (1 min) with increasing concentrations of IgG1, IgG2, and IgG3; (**C**) time-dependent inhibition assays of the three antibodies toward L1, (**D**) residual activity (%) of L1 after preincubation with increasing concentrations of IgG1 (1 min), IgG2 (5 min), and IgG3 (1 min). Each kinetic value is the mean of three different measurements; the standard deviation (SD) was below 2%.

**Figure 6 molecules-25-04453-f006:**
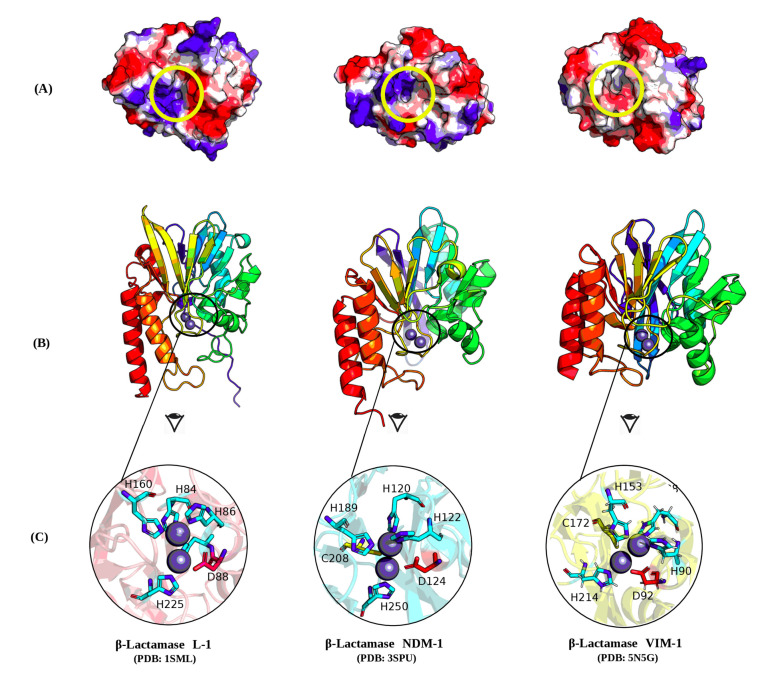
Three-dimensional (3D) structural representation of NDM1, L1, and VIM1 metallo-β-lactamases sharing the same catalytic sites. The crystal structures of NDM1, L1, and VIM1 were extracted from the Protein Data Bank (PDB) using codes 3SPE, 1SML, and 5N5G, respectively. (**A**) Electrostatic potentials are presented on the surface of enzymes using the molecular graphics system PyMOL, with the yellow circle showing a similar conserved cavity forming the catalytic site. (**B**) Enzyme structures with the same orientation are presented in cartoon mode. Similar structural components (sheets and helices) with conserved shapes are highlighted, and rainbow colors code for different structures, highlighting enzyme compartment similarities. (**C**) Catalytic sites for the three enzymes are also highlighted in yellow sticks to show the similarity of these functional regions.

**Table 1 molecules-25-04453-t001:** Proportion of the different IgG fractions purified from 2 mL of hyperimmunized serum.

Fractions	V*Cc (mg)	Proportion of IgGs/Serum	Proportion of IgG Subtypes/Total IgGs
Serum	4	100%	-
IgG1	0.7	17.5%	59.82%
IgG2	0.17	4.25%	14.53%
IgG3	0.3	7.5%	25.64%
		Total = 11.7%	Total = 40.17%

**Table 2 molecules-25-04453-t002:** Half maximal inhibitory concentration (IC_50_) calculated for IgG1, IgG2, and IgG3 versus NDM-1, NDM-1 periplasm, VIM-1, VIM-1 periplasm, L1, and L1 periplasm.

Enzymes	IgG1	IgG2	IgG3
	IC_50_ (μM)	IC_50_ (μM)	IC_50_ (μM)
NDM-1	0.45	>100	15
NDM-1 periplasm	0.42	>100	15
VIM-1	0.18	0.04	0.16
VIM-1 periplasm	0.18	0.036	0.15
L1	0.15	1.0	0.90
L1 periplasm	0.15	0.95	0.90

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
