# Peer review of "Inhibitory Potential of Polyclonal Camel Antibodies against New Delhi Metallo-β-lactamase-1 (NDM-1)"

_molecules, 2020, doi:10.3390/molecules25194453_

Round 1

Reviewer 1 Report

Rahma et al. demonstrated that polyclonal camel antibodies have the potential to inhibit new delhi metallo-Β-Lactamase-1 (Ndm-1). It had nice viewpoints in terms of either preventing or reducing the antibiotic resistance bacteria. The authors also applied camel polycolonal antibodies for NDM-1 that might be able to develop as a kind of treatment. They also showed tridimensional structure of three metallo-β-lactamases NDM1, L1 and VIM1 that recognized by the same IgGs which were sharing the same catalytic sites. It is really wondering why the authors only displayed the results in vitro, without showing any testing result with any bacteria. It doesn’t take time whether or not these antibodies are able to reduce the growth of bacteria having NDM-1. The author should include bacterial testing results even though it would be bad. Or the authors could use bacterial vesicles to know that these antibodies affect the antibiotic resistance. In other case, this manuscript appears to be incomplete.

Author Response

Reviewer #1.

> It is really wondering why the authors only displayed the results in vitro, without showing any testing result with any bacteria. It doesn’t take time whether or not these antibodies are able to reduce the growth of bacteria having NDM-1. The author should include bacterial testing results even though it would be bad. Or the authors could use bacterial vesicles to know that these antibodies affect the antibiotic resistance. In other case, this manuscript appears to be incomplete

We would like to thank the reviewer#1 for the general comments concerning our scientific approach. As suggested, we further investigated whether IgG subclasses are able to reduce the growth of recombinant E. coli BL21(DE3)/codon plus cells containing the recombinant plasmid expressing NDM-1, L1 or VIM-1.  Obtained results showed that the addition of IgG (4 and 8 mg/L) to cell culture was unable to restore the susceptibility of carbapenems. Interestingly, IgGs were able to restore the susceptibility of carbapenems with same IC50, when tested on the periplasm extract of each cultured strain. We deduced that IgGs are able to specifically inhibit the bacteria-expressed enzymes but are unable to translocate through bacterial membrane. This interestingly result could be explained by the high MW of the IgGs, exceeding 100 kDa.

Our result clearly demonstrated the capacity of IgG subclasses to inhibit not only the target NDM-1 but also the VIM-1 and L1 present in the bacterial’s periplasms and emphasis the ongoing investigations on a collection of pathogens having NDM-1, L-1 or VIM-1 MBL release. This result is reported in the discussion section. The text is amended accordingly in red.

Reviewer 2 Report

Dear authors, congrats to the research and for the original idea.

The work is quite interesting due to the potential use of the polyclonal immunotherapy against targets related with a Drug resistance mechanisms, as the beta-lactamases.

The introduction of the manuscript is well written, besides the use of not so recent references. This fact can be caused by the low number of manuscript published with these techniques and with these aims;

The results and the discussion are very clear to demonstrate the immunogenic potential of the NDM enzymatic domain, causing increased production of antibodies, fact very interesting, mainly in these pandemic times, were the same strategy is being studied by several researchers to be used against the COVID-19 virus;

The methodology is adequate and the methods are clear and reproducible, and the in silico approach is very elegant;

The conclusion, besides very short, informs exactly the main points discovered with the paper.

By these facts, I believe that the manuscript can be accepted after few corrections that I will inform follow:

Revise the references: The references listed are not according the guidelines of the molecules, available as a template;

Another point is the necessity of using more recente references to the introduction.

After these few corrections , I will recommend the acceptance.

Author Response

Reviewer #2.

> Revise the references: The references listed are not according the guidelines of the molecules, available as a template

We would like to express thanks to the Reviewer #2 for positively encouraging and supporting our promising results. We review carefully all the references to fit the Molecules Journal instructions and guidelines.

> Another point is the necessity of using more recente references to the introduction.

Recent references (p2, l48) are added to the introduction section and listed with red color into references section.

Reviewer 3 Report

Title: Inhibitory potential of polyclonal camel antibodies against New Delhi metallo-B-lactamase-1 (NDM-1).

This manuscript describes the purification and use of camel antibodies against several clinically relevant MBLs. The authors isolate 3 polyclonal antibodies, one conventional HC/LC antibody (IgG1) and two HC antibodies (IgG2 and 3). Only IgG1 exhibited sub-micromolar activity against NDM-1 while all 3 were reasonably active against VIM-1 (for which a very similar study has already been reported in reference 24) and L1. Although I believe the manuscript is reasonably well written and of potential interest to the readership of Molecules, there are some significant issues in the interpretation of the data that should be resolved.

  1. The reporting of ki by this method is inappropriate. The authors have no idea of the mechanism, yet use an equation for the calculation of ki that is only valid for competitive inhibition (reference 36, line 296). In fact, where such an analysis was conducted on inhibition by a camel antibody of VIM-1 (reference 24), it was found to be predominantly uncompetitive. The long incubation times required to maximize inhibition are consistent with this. An accurate determination of ki is impossible from the data presented and should be removed.
  2. The data in Figures 4 and 5 should include error bars. The pre-incubation times for concentration dependent assays are unclear. On line 124, the authors claim incubation of “IgG1 and IgG2/IgG3 for 1 and 60 min, respectively” while on line 127 they report “NDM-1 and IgG3 were pre-incubated for 30 min.”. The pre-incubation times for Figures 5B and 5D are not reported. I suggest including pre-incubation times in the figure legends.
  3. The discussion of the structural basis of inhibition of the 3 MBLs is not supported by the data. First, there is nothing to suggest that the antibodies bind at the substrate binding site (see point 1). Second, the electrostatic surfaces of the 3 enzymes do not fit the pattern of inhibition. Namely, L-1 and NDM-1 share similar potentials while VIM-1 is quite different. However, all 3 antibodies are better inhibitors of L-1 and VIM-1 than they are of NDM-1 indicating no clear dependence on electrostatic surface potential. I do not see any meaningful conclusions to draw from these structures in the context of the data presented.
  4. Lines 28 and 29 are misquoted from the first sentence of the abstract of reference 1, making the statement completely untrue. Gram-negative infections are most certainly not THE leading cause of mortality and morbidity worldwide.
  5. Typos/misspellings on lines 35 and 36 (“is” should be “are”), 62 (add “bacteria” to end of sentence), 78 (“ten age old” should be “age 10 years”), 120 (“showed” should be “shown”), 192 (“NDM-&-specific”).

Author Response

Reviewer #3.

> The reporting of ki by this method is inappropriate. The authors have no idea of the mechanism, yet use an equation for the calculation of kthat is only valid for competitive inhibition (reference 36, line 296). In fact, where such an analysis was conducted on inhibition by a camel antibody of VIM-1 (reference 24), it was found to be predominantly uncompetitive.

We would like to thank the reviewer#3 for the general comments concerning our scientific approach and suggested modifications. We totally agree the reviewer comment. According to the reviewer recommendation, the reporting of estimated ki has been removed.

For the comments regarding the cited reference data (now reference 27), we accept the critic. However, it is not relevant to compare both antibody models. Indeed those cited in reference 27 are nanobodies with MW of only 15kDa whereas the herein investigated camel IgG1 and HC-IgG2 and -3 have MW of 100 and 150kDa, respectively.

> The long incubation times required to maximize inhibition are consistent with this. An accurate determination of ki is impossible from the data presented and should be removed.

As suggested, data concerning the ki determination have been removed from the text. The enzymatic interaction mechanism will be further investigated. To compare the IgG subclasses’ inhibitory activities, the IC50 values were calculated using increasing concentration of IgGs and after incubation of enzymes/IgGs.

> The data in Figures 4 and 5 should include error bars.

The critic is accepted since it is difficult to see the bar with errors of less than 1%. We have calculated also the standard deviation (SD) which is below 2%.

> The pre-incubation times for concentration dependent assays are unclear. On line 124, the authors claim incubation of “IgG1 and IgG2/IgG3 for 1 and 60 min, respectively” while on line 127 they report “NDM-1 and IgG3 were pre-incubated for 30 min.”

Line 134, the text has been corrected, accordingly.

> The pre-incubation times for Figures 5B and 5D are not reported. I suggest including pre-incubation times in the figure legends.

Pre-incubation times  have been added to the figure 5 legend.

> The discussion of the structural basis of inhibition of the 3 MBLs is not supported by the data. First, there is nothing to suggest that the antibodies bind at the substrate binding site (see point 1). Second, the electrostatic surfaces of the 3 enzymes do not fit the pattern of inhibition. Namely, L-1 and NDM-1 share similar potentials while VIM-1 is quite different. However, all 3 antibodies are better inhibitors of L-1 and VIM-1 than they are of NDM-1 indicating no clear dependence on electrostatic surface potential. I do not see any meaningful conclusions to draw from these structures in the context of the data presented.

We agree with the reviewer#3 comments. We didn’t investigated the IgGs’ binding site which requires more time of investigations using structural modeling and docking softwares. The structural basic study was only carried out to illustrate the general structural topology of the three enzymes, that may support the recorded inhibition of the 3 MBLs. Sufficient information is provided to highlight remarkable concave pockets, conserved between the three enzyme structures which can be involved as patch of interaction on the face of electrostatic potential surfaces (yellow circles).  The observed difference in the electrostatic potentials of the three enzyme surfaces indicated that the molecular interaction is more complex and not only supported by the surface electrostatic potentials. The part (c) of the Figure 6 clarify the lateral chain topology of some crucial MBLs’ amino acids involved in the substrat binding sites, which could be suposed to be involved within the main patch of interaction with IgGs. An advanced structural approach based on modeling and docking will make sens and confirm our preliminary structural observations, as soon as the anti-MBL IgGs amino acid sequences will be available.

> Lines 28 and 29 are misquoted from the first sentence of the abstract of reference 1, making the statement completely untrue. Gram-negative infections are most certainly not THE leading cause of mortality and morbidity worldwide.

We have corrected the sentence that is now: “Infectious diseases caused by Gram-negative bacteria are most certainly cause of morbidity and mortality worldwide” (p1-l33-34).

> Typos/misspellings on lines 35 and 36 “is” should be “are”

Done (p1-l40)

> L62 add “bacteria” to end of sentence

The  Word“bacteria” is added to the end of the sentence (p2-l67)

> L78 “ten age old” should be “age 10 years”

We have replaced “ten age old” by “age 10 years old” (p2-l84)

> L120 “showed” should be “shown”

Done (p4-l127)

> L192 “NDM-&-specific”

We have replaced “new specific antibody inhibitor binders” by “NDM-1 specific binders”(p8-l219)

Round 2

Reviewer 1 Report

It wouldl be sufficient in publishing in this journal.

Author Response

We would like to express thanks to the Reviewer #1 for positively supporting our promising results.

Reviewer 3 Report

The revised manuscript is improved regarding the quality of data content and interpretation. There are still a few minor issues to clear up.

  1. It is not correct to say that IgGs were able to restore susceptibility to carbapenem as is stated in lines 27 and 244. The antibodies are active against the enzymes in the periplasmic extract, but susceptibility indicates the organism is no longer resistant, which is not the case.
  2. I would suggest adding values for IC50 obtained on periplasmic extracts to Table 2. The authors simply state that they get the "same results" as for purified enzyme. I would also remove Table 3, which adds very little.
  3. The revisions incorporated a few additional grammatical errors and typos. I have identified a few, but the manuscript would benefit from a careful proofreading for English. Line 183, "coltures"; line 218-219, I do not understand this sentence; line 247 "interestingly".

Author Response

> It is not correct to say that IgGs were able to restore susceptibility to carbapenem as is stated in lines 27 and 244. The antibodies are active against the enzymes in the periplasmic extract, but susceptibility indicates the organism is no longer resistant, which is not the case.

We would like to thank the reviewer for the general comments and suggested modifications. We totally agree the reviewer comment. According to the reviewer recommendation, the sentences have been amended (p1-l28-29, p9-l248-249).

> I would suggest adding values for IC50 obtained on periplasmic extracts to Table 2. The authors simply state that they get the "same results" as for purified enzyme.

As suggested, all values for IC50 obtained on periplasmic extracts, are added to the Table 2 (p6-l173).

> I would also remove Table 3, which adds very little.

As suggested, Table 3 is removed from the text.

> The revisions incorporated a few additional grammatical errors and typos. I have identified a few, but the manuscript would benefit from a careful proofreading for English.

A careful proofreading for English is done.

>Line 183, "coltures";

The word has been corrected (p7-l186)

>line 218-219, I do not understand this sentence

The sentence has been amended (p8-l222-223)

> line 247 "interestingly".

Done